# Acoustic Anomalies and the Critical Slowing-Down Behavior of MAPbCl_3_ Single Crystals Studied by Brillouin Light Scattering

**DOI:** 10.3390/ma15103692

**Published:** 2022-05-21

**Authors:** Jeong Woo Lee, Furqanul Hassan Naqvi, Jae-Hyeon Ko, Tae Heon Kim, Chang Won Ahn

**Affiliations:** 1School of Nano Convergence Technology, Nano Convergence Technology Center, Hallym University, Chuncheon 24252, Korea; dlwjddn1992@naver.com (J.W.L.); furqanhassan05@gmail.com (F.H.N.); 2Department of Physics and Energy Harvest-Storage Research Center (EHSRC), University of Ulsan, Ulsan 44610, Korea; thkim79@ulsan.ac.kr (T.H.K.); cwahn@ulsan.ac.kr (C.W.A.)

**Keywords:** halide perovskite, Brillouin scattering, elastic constant, sound velocity

## Abstract

Inelastic light scattering spectra of organic–inorganic halide perovskite MAPbCl_3_ single crystals were investigated by using Brillouin spectroscopy. Sound velocities and acoustic absorption coefficients of longitudinal and transverse acoustic modes propagating along the cubic [100] direction were determined in a wide temperature range. The sound velocities exhibited softening upon cooling in the cubic phase, which was accompanied by the increasing acoustic damping. The obtained relaxation time showed a critical slowing-down behavior, revealing the order–disorder nature of the phase transition, which is consistent with the growth of strong central peaks upon cooling toward the phase transition point. The temperature dependences of the two elastic constants *C*_11_ and *C*_44_ were obtained in the cubic phase for the first time. The comparison of *C*_11_ and *C*_44_ with those of other halide perovskites showed that *C*_11_ of MAPbCl_3_ is larger and *C*_44_ is slightly smaller compared to the values of MAPbBr_3_ and MAPbI_3_. It suggests that MAPbCl_3_ has a more compact structure (smaller lattice constant) along with stronger binding forces, causing larger *C*_11_ and bulk modulus in this compound, and that the shear rigidity is exceedingly small similar to other halide perovskites. The reported elastic constants in this study may serve as a testbed for theoretical and calculational approaches for MAPbCl_3_.

## 1. Introduction

Perovskites with a generic ABX_3_ structure are being employed in several research fields [1,2]. Hybrid organic–inorganic halide perovskites (HOPs) have arisen as key materials for application in optoelectronic and photovoltaic devices [3,4], such as light-emitting diodes [5], photodetectors [6] and, especially, solar cells [7], which have shown a drastic increase in the solar power conversion efficiency from 3.9% to 25.7% in a few years [8]. Various experimental and theoretical approaches have been adopted to understand the structural properties, phase transition behaviors, and device performances of HOPs [9]. However, a deeper understanding of the fundamental intrinsic properties and phase transition behaviors of these materials are prerequisites to further improvements of the devices based on HOPs. In this context, single crystals may be ideal for deeper investigation from a fundamental point of view due to the lack of any effects from grain boundaries, morphologies, and defects. Single crystals can be made using a variety of techniques, including solvent evaporation, temperature lowering, inverse crystallization, and antisolvent vapor-assisted methods [10]. We chose the solvent evaporation method among these to synthesize single crystals of high optical quality.

One of the fundamental properties of solids is their elastic properties, including various elastic moduli and acoustic absorption coefficients [11,12]. Elastic constants are directly related to the interatomic forces in the crystal and provide an experimental platform for testing theoretical/calculational models. There are several experimental methods for probing these properties, such as low-frequency indentation and ultrasonic methods [13], etc. Among them, Brillouin light scattering is a nondestructive, noncontact method that is a powerful tool in the determination of the elastic constants, sound velocities, and acoustic attenuation coefficients of acoustic waves propagating in condensed matters [14,15,16,17,18,19,20]. Despite extensive studies on HOPs in terms of various experimental techniques, there are only a few acoustic studies on these materials [13,21,22,23,24,25,26,27]. Various acoustic properties of CH_3_NH_3_PbX_3_ (MAPbX_3_ with X = Cl, Br, I) and CH(NH_2_)_2_PbX_3_ (FAPbX_3_ with X = Cl, Br, I) have been reported at room temperature or as a function of temperature. In addition, there are a few calculation studies on the mechanical properties of these compounds [12,28].

Ferreira et al. used coherent inelastic neutron scattering and Brillouin scattering techniques to probe the elastic properties of several lead-based halide perovskites [24]. They found a very low shear modulus and softness in these compounds. Significant elastic anisotropy was confirmed in MAPbBr_3_ measured by the laser ultrasonic technique [13]. Harwell et al. used resonant ultrasound spectroscopy to investigate the temperature dependence of acoustic resonant frequencies and the acoustic losses of methylammonium lead halide perovskites [25]. They found that acoustic losses are substantially large in the tetragonal phase of MAPbBr_3_ and MAPbI_3_, which was attributed to the mobility of the ferroelastic twin walls. The overall changes in the elastic properties were explained in terms of the MA order and disorder in each phase [25]. Zhevstovskikh et al. used the ultrasonic technique to study the changes in the sound velocity and the attenuation coefficient of both the longitudinal and transverse acoustic waves of MAPbI_3_ single crystals in the MHz range [27]. This study revealed a step-like change in the sound velocity and a sharp attenuation peak near the tetragonal–orthorhombic phase transition point. These anomalies were interpreted based on the phenomenological theory of the order parameter–strain coupling and the fluctuations of the order parameter.

Another interesting study on the high-frequency dynamics of MAPbX_3_ was reported by Anusca et al. [23]. They showed that a fast relaxation process appears in the GHz range probed by broadband dielectric spectroscopy, the relaxation time of which exhibited a critical slowing-down behavior near the phase transition point, indicating the order–disorder nature of the phase transition. They also reported significant changes in the sound velocity and the attenuation coefficient at both phase transitions of the cubic-to-tetragonal and tetragonal-to-orthorhombic phases. Since the GHz relaxation process is exactly overlapped with the frequency window of the Brillouin spectroscopy, it would be interesting to find out the correlation between the dielectric relaxation process and the Brillouin scattering results.

The information on the elastic properties of halide perovskites is important for the investigation of their stability in device applications or material preparation. The purpose of this study is to investigate the temperature dependence of the acoustic properties of MAPbCl_3_ single crystals in the hypersonic range by using Brillouin spectroscopy. Especially, the temperature dependence of some elastic constants will be reported for the first time and their correlation with the phase transition will be discussed in detail. MAPbCl_3_ is the most stable halide perovskite, with a bandgap energy of ~2.9 eV [29]. It has been studied in terms of various experimental methods [30,31,32], but the acoustic study is very rare [23,25]. The obtained results will be compared to previous studies, especially the high-frequency dielectric properties reported by Anusca et al. [23].

## 2. Materials and Methods

Lead chloride (PbCl_2_, 99.999%), methylamine (CH_3_NH_2_, 40% in water), hydrochloric acid (HCl, 37%, ACS reagent), dimethyl sulfoxide (DMSO, anhydrous ≥ 99.9%), diethyl ether (HPLC grade, ≥99.9%), and ethanol (anhydrous 99.5%) were purchased from Sigma Aldrich (St. Louis, MO, USA). Methylammonium chloride (MACl) was synthesized by dropwise addition of HCl into ice-bath-cooled methylamine in the 1.2:1 molar ratio followed by stirring until complete dissolution. The obtained solution was then dried into a rotary evaporator at 60 °C under a vacuum. After 3 h, the solution had completely dried, and a white shiny MACl powder was obtained. MACl powder was then purified by dissolving it in ethanol (200 mL) at 40 °C. Next, diethyl ether (200 mL) was added to achieve precipitation in this solution. The purification step was repeated twice. Finally, the obtained powder was dried overnight in a vacuum oven at 60 °C.

Equimolar solutions of the obtained white MACl powder (0.01 M) and PbCl_2_ (0.01 M) were dissolved in DMSO (10 mL) by stirring at 60 °C. After complete dissolution, the solution was filtered through a 0.22 µm syringe filter into a crystallization dish. The dish was covered with aluminum foil and left undisturbed for 1–2 days at a constant temperature of 100 °C. After 1–2 days, transparent MAPbCl_3_ crystals with a typical size of 5 × 5 × 2 mm^3^ were obtained. Afterward, the crystals were washed with acetone and dried overnight in a vacuum oven at 60 °C. Figure 1a shows the photo of the grown single crystal which was used for the Brillouin measurement.

A high-resolution powder X-ray diffractometer (PANalytical X’pert Pro MPD, Malvern, UK) was used to record X-ray diffraction (XRD) patterns at room temperature, with Cu Kβ radiations with λ = 1.5406 Å as an excitation source to probe the crystal at a scan rate of 3° per minute in a scan range of 10° < 2θ < 60°.

Brillouin spectra were measured by using a typical tandem multi-pass Fabry–Perot interferometer (TFP-2, JRS Co., Zürich, Switzerland). The free spectral range was 50 GHz, and the scan range was ±35.7 GHz. The typical finesse of the interferometer was around 100. A modified microscope (BH-2, Olympus, Tokyo, Japan) including a small prism for redirecting the laser beam, was used for the backscattering experiment. A diode-pumped solid-state single-mode laser (Excelsior 532−300, Spectra Physics, Santa Clara, CA, USA) at a wavelength of 532 nm was used as an excitation light source. A conventional photon-counting system combined with a multichannel analyzer (1024 channels) was used to detect and average the signal. The sample was put in a cryostat stage (THMS 600, Linkam, Tadworth, UK) for temperature control. All the measurements were taken in the temperature range from −196 °C to 40 °C.

## 3. Results and Discussion

The powder X-ray diffraction for the obtained single crystal, which is shown in Figure 1b, was consistent with previous reports [33]. Figure 2a shows a temperature dependence of the Brillouin spectrum of the MAPbCl_3_ single crystal. In the high-temperature cubic phase, the spectrum consists of two Brillouin doublets appearing at ~25 GHz and ~8 GHz. The high- and the low-frequency peaks correspond to the longitudinal acoustic (LA) and the transverse acoustic (TA) modes, respectively. As temperature decreases, a quasi-elastic central peak begins to appear at approximately −80 °C and grows upon further cooling. This is evident from the intensity color plot, as shown in Figure 2b. The LA mode is split at about −116 °C, which corresponds to the tetragonal-to-orthorhombic phase transition point. The TA mode is split as well at the same temperature, which is, however, not clearly seen in Figure 2a.

For quantitative analysis, the measured spectrum was curve-fitted by using a Voigt function. It is a convolution function of the Lorentzian function (an approximate response function of the damped harmonic oscillator) and the Gaussian instrumental function. The acoustic-mode frequency (*ν*_B_) and the full width at half maximum (FWHM, Γ_B_) was derived as a function of temperature. Figure 3a,b show the temperature dependences of the *ν*_B_ and the Γ_B_ of the LA mode, respectively, for the cooling process. The temperature dependence of the *ν*_B_, as well as its splitting at −116 °C, was reproducible by subsequent measurements. MAPbCl_3_ has been known to undergo two structural phase transitions, from the cubic-to-tetragonal phase at approximately −95 °C (=*T*_C-T_), and then from the tetragonal-to-orthorhombic phase at about −116 °C (=*T*_T-O_). The LA mode shows gradual softening and then a slight upward hardening upon cooling in the cubic phase. The *ν*_B_ exhibits a small cusp, while the Γ_B_ displays a clear maximum near *T*_C-T_. Appendix A show the comparison of the Brillouin data and the real part of the dielectric permittivity reported in [34]. the *T*_C-T_ is characterized by the maxima of both the permittivity and the Γ_B_, while the permittivity and the *ν*_B_ drop precipitously at *T*_T-O_. These comparisons indicate that the acoustic anomalies are strongly correlated with the changes in the dielectric properties.

The Brillouin spectra were recorded upon heating as well. The comparison between cooling and heating processes is shown in Appendix A. The splitting of the LA mode at low temperatures and the softening of the LA mode in addition to a significant damping factor in the cubic phase were confirmed in the heating process. However, the data are a little bit scattered, which may be probably due to the sample degradation caused by the long exposure to the ambient condition.

Both LA- and TA-mode frequencies are split at *T*_T-O_, as described above. Especially, the Γ_B_ of the LA mode exhibits a sharp damping peak at *T*_T-O_. The half-widths are small in the orthorhombic phase, while the mode frequencies show slight hardening upon further cooling below *T*_T-O_. There are small but noticeable anomalies near −140 °C, the origin of which is not clear. However, previous acoustic studies by resonant ultrasound spectroscopy revealed a clear acoustic damping peak in the same temperature range, which was tentatively attributed to some freezing process of the MA rotation and/or the hopping of Cl ions between vacancies [25]. However, this small anomaly does not appear during the heating process (Appendix A), indicating that the relevant structural change may be metastable.

Figure 4 shows the temperature dependence of the *ν*_B_ of the TA mode for the cooling process. It displays a gradual softening upon cooling in the cubic phase, a broad minimum at *T*_C-T_, and a sudden increase along with splitting at *T*_T-O_. The Γ_B_ of the TA mode, which is shown in Appendix A, is rather scattered but shows a small peak near *T*_T-O_. Splitting of the TA mode at and below *T*_T-O_ is seen and is very similar to the case of the LA mode. The splitting of the acoustic-mode frequencies was confirmed in MAPbBr_3_ as well and ascribed to the presence of multiply-strained domains [22]. In this case, there may exist several multiply-strained domains in the focal point of the probe laser beam, where differently strained areas with different refractive indices would give rise to split Brillouin doublets. Therefore, a quantitative analysis for the orthorhombic phase was difficult to carry out. This result, combined with [22], indicates that the formation of multiply-strained domains in the low-temperature orthorhombic phase seems to be a common phenomenon in MA-based halide perovskite single crystals.

The data of *ν*_B_ and Γ_B_ were transformed into the sound velocity *V* and the acoustic absorption coefficient *α* via the following equations valid for the backscattering geometry;
(1)$$V=\frac{\nu_{B}\lambda }{2n},$$
(2)$$\alpha =\frac{\pi \Gamma_{B}}{V}.$$

In these equations, *λ* is the laser wavelength (=532 nm) and *n* is the refractive index. Figure 5a,b show the temperature dependence of the sound velocity and the absorption coefficient of the LA mode, respectively. The refractive index of *n* = 1.9 at 532 nm, reported in the previous study [35], was used for the calculation. The temperature dependence of *n* was not considered. The longitudinal sound velocity is approximately 3599 m/s at room temperature and shows a mild softening upon cooling. This elastic softening is accompanied by a drastic increase in the absorption coefficient, a more than five times increase upon the change of *T* from room temperature to *T*_C-T_. It indicates that the longitudinal sound wave is strongly coupled to other degrees of freedom associated with the cubic-to-tetragonal phase transition and that the dissipative energy exchange between them becomes very active near *T*_C-T_. There are two reports on the longitudinal sound velocities of MAPbCl_3_ [23,26]. The longitudinal sound velocity of ceramic MAPbCl_3_ is approximately 3450 m/s at 0 °C [23], which is comparable to the result of this study. However, the other result on the single-crystal MAPbCl_3_ shows that the sound velocity is 4000 m/s at room temperature [26], which is much larger than the other two results, including the present study. The sound velocity of the TA mode is shown in Figure 6. It decreases from ~1098 m/s to ~967 m/s at *T*_C-T_ and then increases in the tetragonal phase drastically. The sound velocity exhibits two branches in the orthorhombic phase corresponding to the two acoustic modes (split TA mode) shown in Figure 4. The transverse sound velocity of this study is much smaller compared to the previous report of 1770 m/s [26]. Since the phonon propagation direction was not completely determined and the laser wavelength was different in the previous work [26], it is not clear whether the difference in the sound velocities is due to the dispersion effect or the difference in the phonon direction.

The coupling between the strain caused by the LA mode and the other degrees of freedom relevant to the phase transition is more clearly seen from the elastic constant data. MAPbCl_3_ maintains a cubic symmetry at high temperatures down to ~ *T*_C-T_, at which the cubic phase changes into a tetragonal one. There are three independent elastic constants in the cubic phase, namely *C*_11_, *C*_12_, and *C*_44_ [14]. Since the phonon propagation direction of the present experiment is the [100] direction, the LA and TA modes shown in Figure 2 correspond to the *C*_11_ and *C*_44_, respectively. The reported density of *ρ* = 3171 kg/m^3^ [36] was used to obtain the two elastic constants as *ρ**V*^2^ in the cubic phase. Table 1 shows the comparison of the sound velocities and the two elastic constants of the MA-based halide perovskites at room temperature. There is no experimental result for the elastic constant of MAPbCl_3_. Only theoretical calculations provided the estimations for the elastic constants [12,28]. The reported values for *C*_11_ from these theoretical studies are 39.5 and 42.1 GPa, which are very similar to the present result of 41.0 GPa. *C*_44_ has been theoretically estimated to be 2.9 or 6.5 GPa, and our result of 3.74 GPa is located in this range. These consistent results suggest both theoretical studies are reliable, although the difference in predicted *C*_44_ values is rather large, and that our experimental results provide a reliable testbed for further theoretical works.

Figure 7 shows the change in the elastic constants as a function of the lattice constant (and, thus, X in MAPbX_3_) in lead-based halide perovskites. Since the elastic bulk modulus cannot be obtained from the present result due to the lack of *C*_12_, theoretical values of [28] were included for comparison. The *C*_11_ and the bulk modulus increase with the decreasing lattice constant, while *C*_44_ values are relatively similar and very small. A more compact structure at a smaller lattice constant is responsible for the larger strength of the binding interactions, which was attributed to the more symmetric MA cations and the resulting steric effect [24]. On the other hand, the exceedingly small *C*_44_ values of the three compounds indicate their very unstable structure against the shear forces, resulting in large elastic anisotropy and an exceptionally low shear modulus.

The temperature dependences of the obtained elastic constants in the cubic phase are shown in Figure 8. Both elastic constants display softening upon cooling toward *T*_C-T_. *C*_11_ shows a maximum near ~0 °C and begins to be softened below this temperature, which indicates that the coupling of the LA mode and other degrees of freedom begins near this temperature. *C*_11_ changes from ~41.2 GPa at about 0 °C to ~40.4 GPa at −90 °C. In the case of *C*_44_, it continuously decreases from ~3.82 GPa at 40 °C to ~2.96 GPa at *T*_C-T_. The surprisingly small *C*_44_ was also observed from other halide perovskites, such as MAPbBr_3_ [13,22]. The *C*_44_ of MAPbBr_3_ is around 3.5 GPa at room temperature and shows the smallest value along the [100] direction in the slowness curve [13]. The *C*_44_ of MAPbI_3_ is reported to be approximately 7.3 GPa [24] or 4.8 GPa [28]. On the other hand, the FA-based halide perovskites exhibit even smaller *C*_44_ values than MA-based materials [24]. It indicates that the extremely low shear rigidity may be a common characteristic in MA- and FA-based halide perovskites, with the latter being softer to the shear stress.

The above results clearly showed that the acoustic mode is coupled to other degrees of freedom, which are associated with the cubic-to-tetragonal phase transition, in the cubic phase. Létoublon et al. suggested that the disordered reorientational motions of the MA cations are responsible for the translation–rotation coupling, which induces the elastic softening [22]. They revealed that the low-frequency Brillouin spectra of MAPbBr_3_ are characterized by the central peak coupled to the LA mode, which is another indication of the order–disorder characteristic of the cubic-to-tetragonal phase transition. In another work, the rotations and tilts of the corner-sharing oxygen octahedra were suggested to be the origin of the acoustic softening [13]. Especially, the rotational motions may be responsible for the extremely low shear modulus. We assume that the translation–rotation coupling between the strains caused by the acoustic modes and the relevant degrees of freedom can be explained in terms of the phenomenological approach based on the Ginzburg–Landau free energy expansion [37]. In this model, which can be applied to both cases of linear and quadratic couplings, the relaxation process coupled to the strain is dynamic and has frequency-dependent properties. The relaxation time of the relevant process that couples to the acoustic waves can quantitatively be analyzed based on an assumption of a single relaxation process. This assumption may be justified in the high-temperature range, where the distribution of relaxation times is expected to be narrow. This narrow distribution is supported by the previous results from broadband dielectric spectroscopy [23]. The relaxation time $\tau_{LA}$ is expressed by the following equation [38], which was derived from a simple acoustic dispersion relationship;
(3)$$\tau_{LA}=\frac{\Gamma_{B}-\Gamma_{\infty }}{2\pi \left(\nu_{\infty }^{2}-\nu_{B}^{2}\right)}$$

Here, $\nu_{\infty }$ and $\Gamma_{\infty }$ are an unrelaxed Brillouin shift at the high-frequency limit and the high-frequency background damping, respectively. Both represent the quantities that are not related to the softening of the LA mode and the local or macroscopic phase transition.

The widest measurement temperature range covered by this study makes it possible for us to carry out a quantitative analysis for the LA modes. The $\nu_{\infty }$ and $\Gamma_{\infty }$ were derived from the high-temperature values of the $\nu_{B}$ and the Γ_B_. The calculated relaxation times are shown in Figure 9. The $\tau_{LA}$ increases from ~16 ps to ~24 ps when the temperature decreases from −60 °C to −100 °C. The orders of magnitude of $\tau_{LA}$ are consistent with the value of 28 ps derived from the central peak reported in the previous work [19]. The breakpoint of approximately −100 °C is nearly the same as, but slightly lower than, *T*_C-T_. We could observe the central peak as well, but the quantitative analysis was difficult to carry out due to the strong Rayleigh line, as described below.

If the relevant degrees of freedom coupled to the acoustic modes are relaxation processes, they can be probed in terms of broadband dielectric spectroscopy as well. Anusca et al. reported broadband dielectric data where they could observe strong dielectric relaxation in the GHz range [23]. The obtained relaxation time exhibited a critical slowing-down behavior, which was attributed to the rotational motions of the MA cations [23]. Figure 9 includes the dielectric relaxation time reported in this reference. Both acoustic and dielectric relaxation times exhibit quite similar temperature dependences near *T*_C-T_, i.e., both values increase (slow-down) toward this transition temperature upon cooling in the cubic phase. This result clearly shows that the microscopic origin of both relaxation processes, acoustic and dielectric, is the same, i.e., the correlated MA motions [23], which accompany both dipole changes under an oscillating electric field and polarization fluctuations.

The linear temperature dependence of the relaxation time is a typical characteristic of the critical slowing-down behavior. The two linear parts in Figure 9 were fitted in terms of the following equation [39]:(4)$$\frac{1}{\tau_{LA}}=\frac{1}{\tau_{0}}\left(\frac{T-T_{0}}{T_{0}}\right)$$

The obtained fitting parameters are summarized in Table 2. This kind of critical slowing-down behavior indicates that the nature of the phase transition is of an order–disorder type; thus, the relaxation time increases upon approaching the phase transition point due to the growing volume (or correlation length) of the correlated MA cations.

Yamamuro et al. discussed the nature of the cubic-to-tetragonal phase transition from their calorimetric, infrared, and dielectric measurements [40,41], according to which the MA ions are suggested to be located in equivalent states in the high-temperature prototype phase. The phase transition is accompanied by a partial ordering of the disordered MA configurations in the cubic phase. The estimated entropy indicates that the phase transition is of the order–disorder type, which is consistent with the critical slowing-down behavior observed in this study and the dielectric study [23]. Similar behaviors were observed from other systems, such as ferroelectric BaTiO_3_ [36] and relaxors [42], etc., where nanoscale precursor polar clusters or polar nanoregions grow upon cooling toward the transition temperature or freezing temperature.

The fast relaxation process is in many cases responsible for the formation of quasi-elastic central peaks in the inelastic light scattering spectrum. A low-frequency Raman study on MAPbBr_3_ revealed the existence of the central peak centered at a zero frequency [22]. Brillouin spectra also exhibit a low-frequency quasi-elastic central peak, as shown in Figure 10a, which was measured in a wide-frequency range of ±546.86 GHz (±18.26 cm^−1^). The central peak grows significantly as temperature decreases but becomes very small below *T*_T-O_, as is revealed in Figure 10b. The central peak was curve-fitted by using the convolution of the Lorentzian function centered at a zero frequency (which represents a relaxation process of a Debye type) and the Gaussian instrumental function. The temperature dependences of the intensity and the FWHM of the central peak are plotted in Figure 10b. The intensity is large in the cubic phase, decreases significantly in the tetragonal phase, and then becomes very small in the orthorhombic phase. The drastic change in the intensity indicates that the relaxation process is very active in the cubic phase. The quantitative analysis of the relaxation time was difficult due to the strong Rayleigh peak, which distorts the low-frequency central peak. However, an approximate relaxation time $\tau_{CP}$ can be estimated from the FWHM via $\tau_{CP}=1/\pi \Gamma_{CP}$, where $\Gamma_{CP}$ is the FWHM of the central peak. Figure 10b shows that $\tau_{CP}$ decreases from ~1 ps to 3.2 ps upon cooling in the cubic phase. The general behavior mimicking the slowing-down behavior is similar to that calculated from the LA mode, but the absolute values are one order of magnitude smaller, which may be due to the insufficient spectral range, especially the low-frequency spectral features, which were distorted by the strong Rayleigh peak.

The present results show clear evidence for the existence of precursor dynamics of MAPbCl_3_ in the prototype cubic phase. The correlated MA regions couple to the acoustic waves, interact with them, and induce acoustic softening as well as the formation of a strong central peak near the phase transition point. The estimated relaxation time from the LA-mode anomalies follows the critical slowing-down behavior, which demonstrates the order–disorder nature of the corresponding phase transition. The first and detailed elastic constant data can be served as an experimental testbed for checking the reliability of theoretical approaches to this important class of materials.

## 4. Conclusions

The acoustic properties of MAPbCl_3_ single crystals were investigated in a wide temperature range from 40 °C to −196 °C by using Brillouin spectroscopy. The sound velocities and the absorption coefficients of both the longitudinal and transverse acoustic modes propagating along the cubic [100] direction were obtained as a function of temperature, and the temperature dependences of the two elastic constants, *C*_11_ and *C*_44_, could be obtained in the cubic phase for the first time. The two-phase transition points were clearly identified from the anomalous changes in the sound velocities and absorption coefficients. The low-temperature orthorhombic phase was characterized by split acoustic modes, which were attributed to the presence of multiply-strained domains. The *C*_11_ (=41.0 GPa) of MAPbCl_3_ was larger than those of MAPbBr_3_ and MAPbI_3_, indicating a more compact crystal structure and stronger binding interactions, while the *C*_44_ (=3.74 GPa) value was exceedingly small, similar to other MA- and FA-based halide compounds. The anomalous changes in the acoustic properties indicated the coupling of the acoustic waves to the dynamic MA cations and their collective rotational motions in the prototype cubic phase. The obtained relaxation time followed the critical slowing-down behavior consistent with previous dielectric studies demonstrating that the cubic-to-tetragonal phase transition is of an order–disorder type. This is consistent with the observation of strong central peaks near the phase transition point, reflecting the existence of an active relaxation process, which can be probed by dielectric spectroscopy and coupled to the acoustic waves.

## Figures and Tables

**Figure 1 materials-15-03692-f001:**
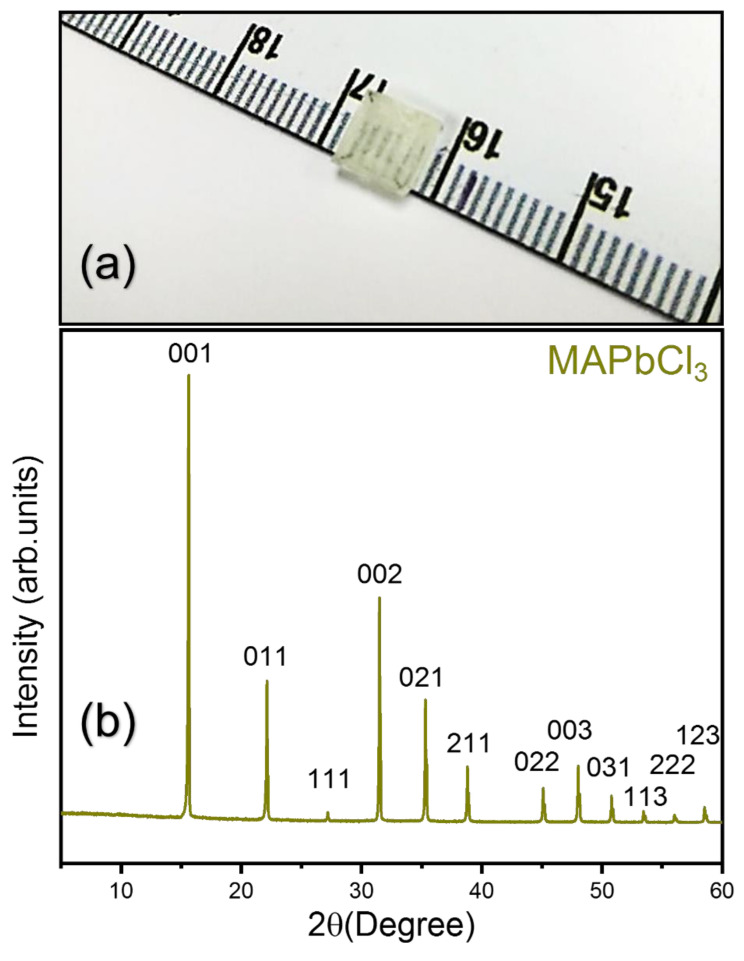
(Color online) (**a**) A picture of the grown single crystal. (**b**) The X-ray diffraction pattern of the powder MAPbCl_3_.

**Figure 2 materials-15-03692-f002:**
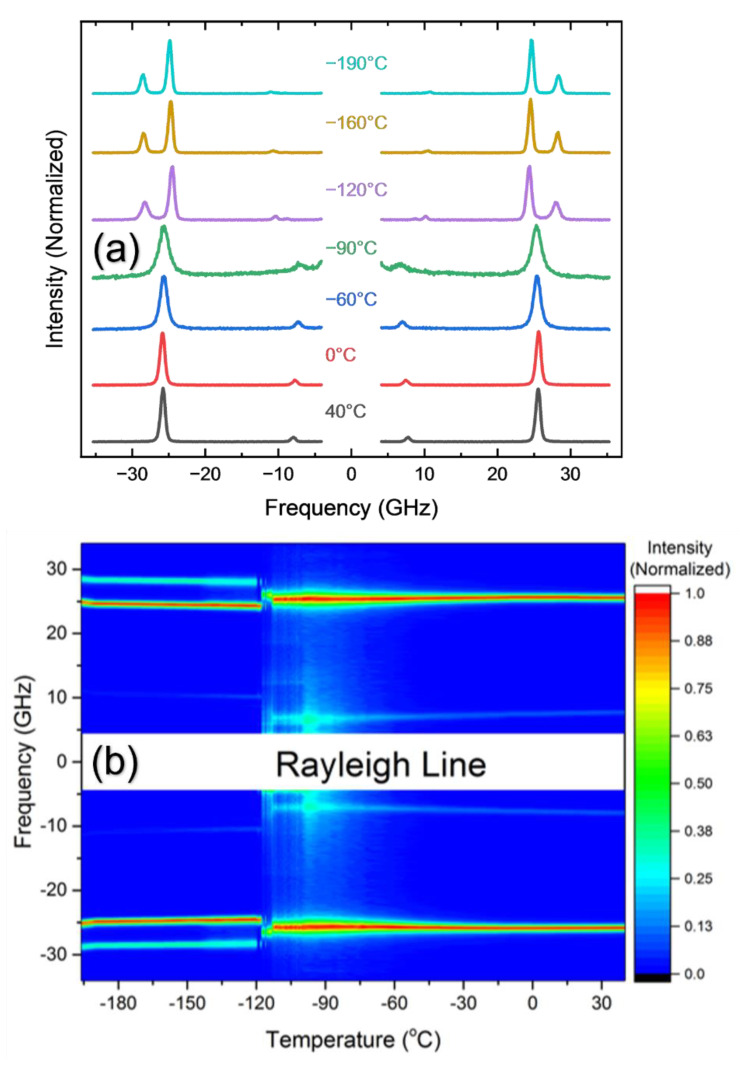
(Color online) (**a**) Temperature dependence of the Brillouin spectrum and (**b**) the intensity color plot of the measured spectra.

**Figure 3 materials-15-03692-f003:**
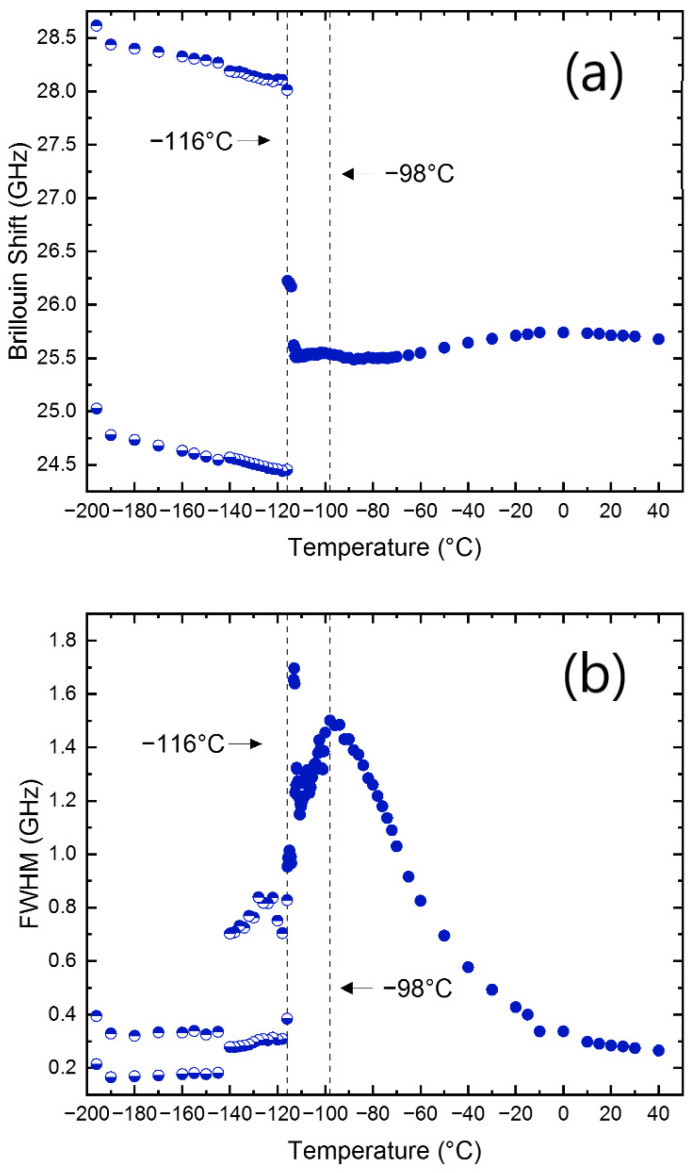
(Color online). Temperature dependences of (**a**) the Brillouin frequency shift and (**b**) the FWHM of the LA mode propagating along the [100] direction, measured upon cooling. The half-filled circles represent the split peaks. The two symbols at low temperatures indicate the data from the split LA mode.

**Figure 4 materials-15-03692-f004:**
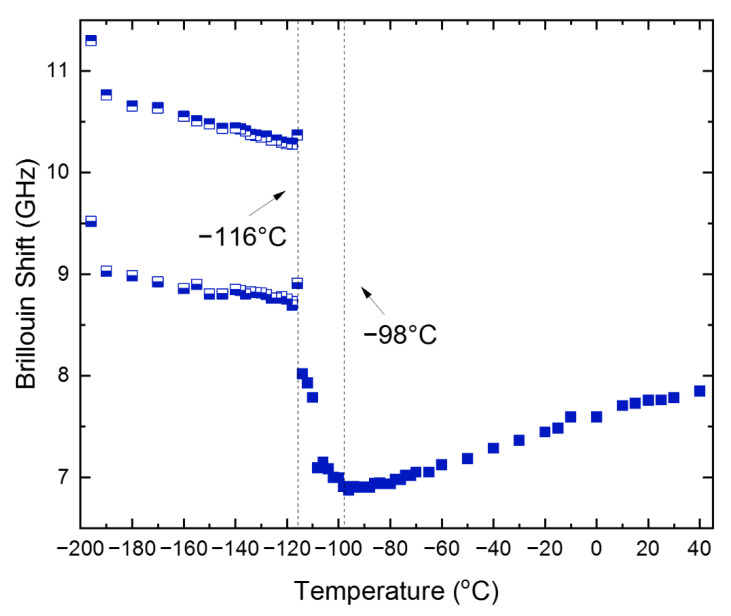
(Color online). Temperature dependences of the Brillouin frequency shift of the TA mode propagating along the [100] direction, measured upon cooling. The two symbols at low temperatures indicate the data from the split TA mode.

**Figure 5 materials-15-03692-f005:**
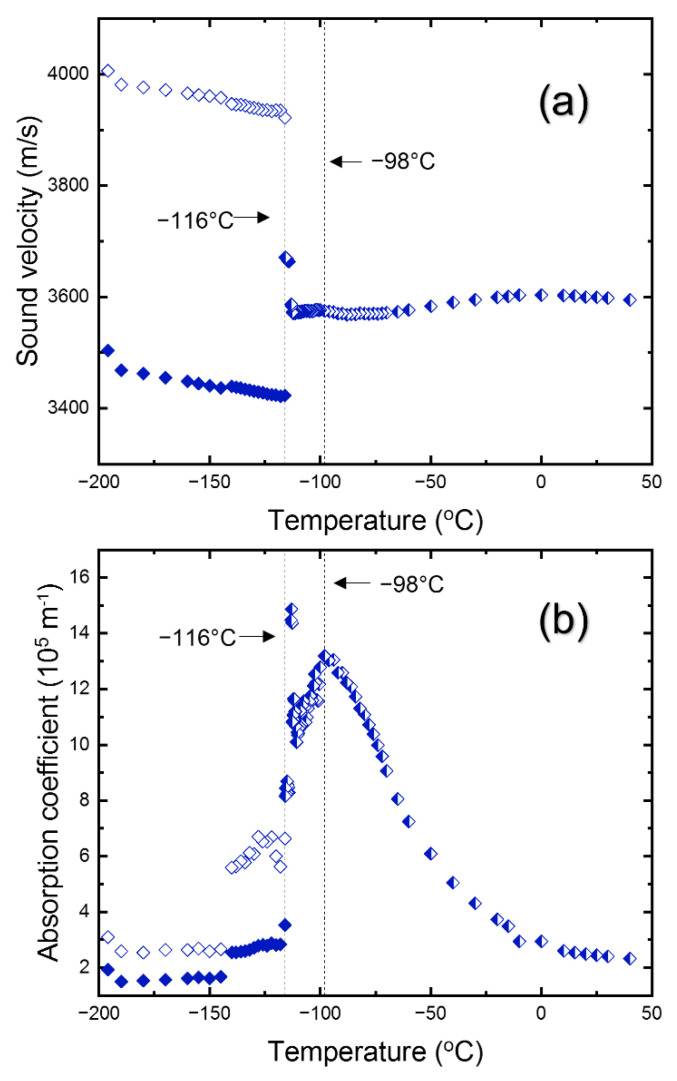
(Color online). Temperature dependences of (**a**) the sound velocity and (**b**) the absorption coefficient of the LA mode. Phase boundaries are shown as vertical dotted lines along with the phase transition temperatures. The two symbols at low temperatures indicate the data from the split LA mode.

**Figure 6 materials-15-03692-f006:**
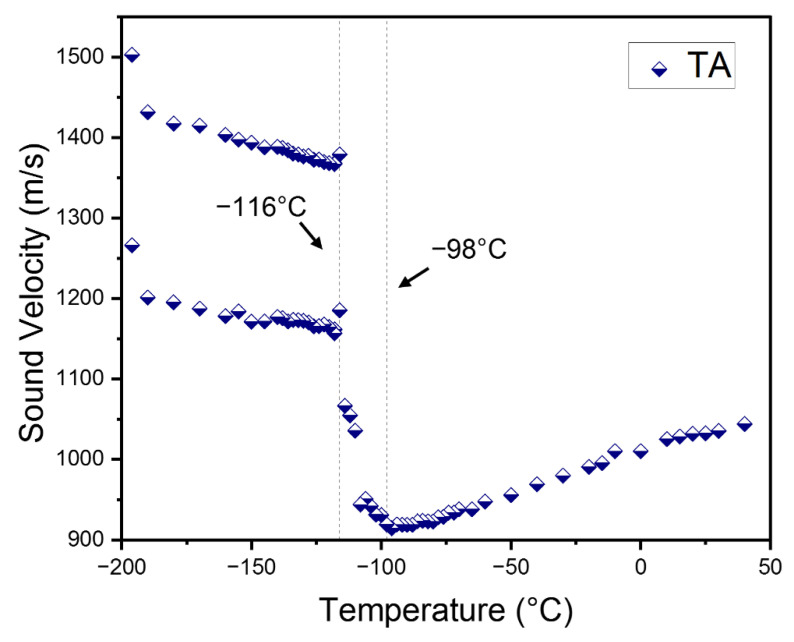
(Color online). Temperature dependences of the sound velocity of the TA mode. Phase boundaries are shown as vertical dotted lines along with the phase transition temperatures.

**Figure 7 materials-15-03692-f007:**
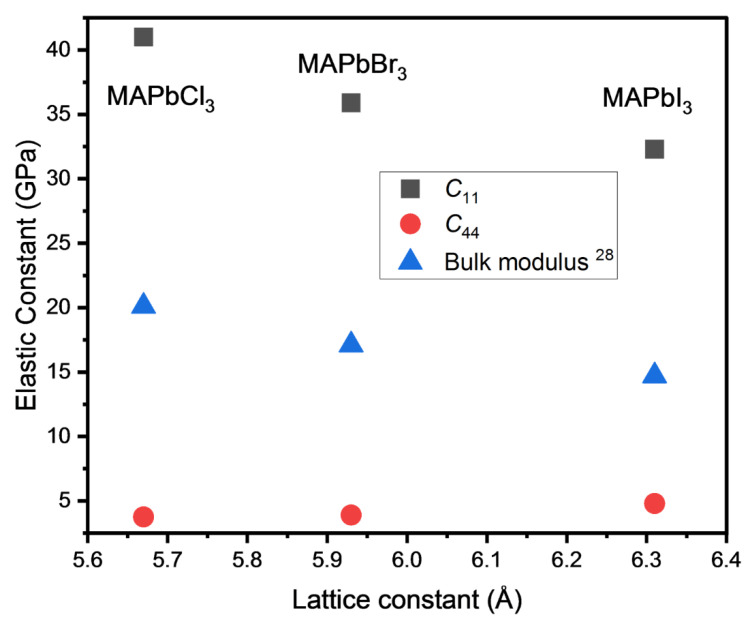
(Color online). Comparison of two elastic constants, *C*_11_ and *C*_44_, and the bulk modulus of three MA-based halide perovskites at room temperature. The *C*_11_ and *C*_44_ of MAPbCl_3_ are from the present work and other values are from several references, as shown in Table 1.

**Figure 8 materials-15-03692-f008:**
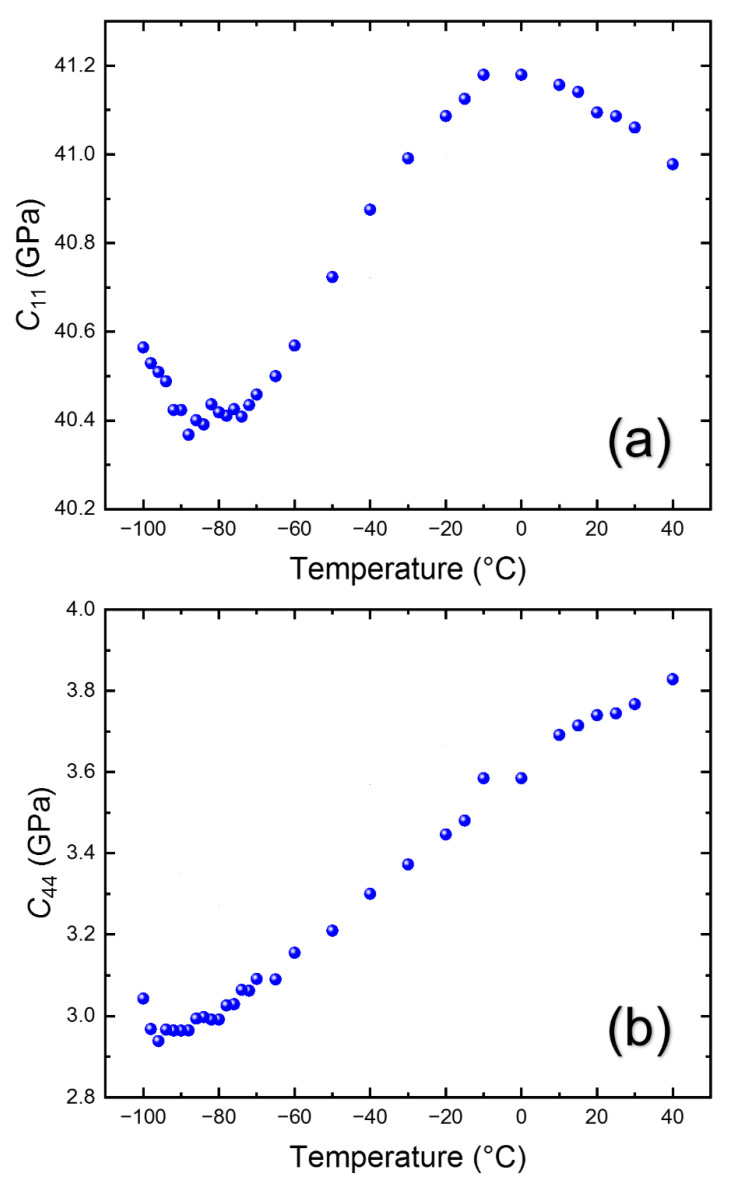
(Color online). Temperature dependences of (**a**) *C*_11_ and (**b**) *C*_44_ in the cubic phase.

**Figure 9 materials-15-03692-f009:**
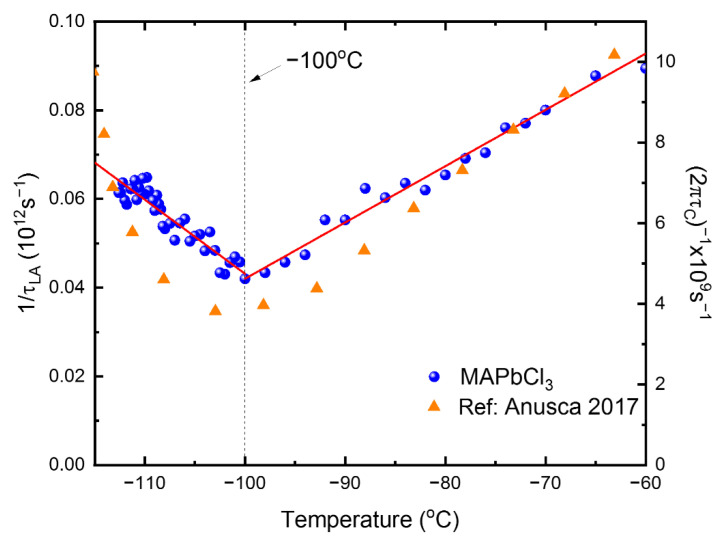
(Color online). Temperature dependence of the inverse relaxation time obtained by using the acoustic anomalies of the LA mode and Equation (3) [20]. The solid line is the best-fitted result by using Equation (4). The inverse dielectric relaxation time *τ*_c_ taken from [23] is shown for comparison.

**Figure 10 materials-15-03692-f010:**
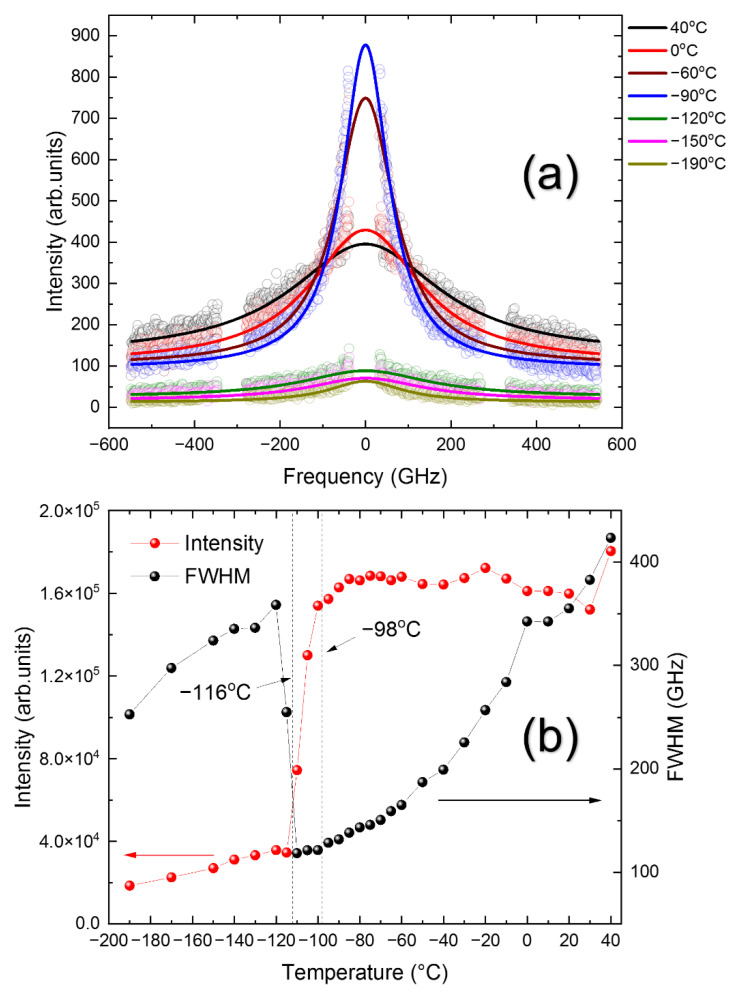
(Color online). (**a**) The central-peak spectra at several temperatures and (**b**) the temperature dependence of the central-peak intensity and its half-width.

**Table 1 materials-15-03692-t001:** Comparison of sound velocities and elastic constants of several halide perovskites.

Parameters	MAPbCl_3_ (This Work)	MAPbCl_3_ (Other Works)	MAPbBr_3_ (Other Works)	MAPbI_3_ (Other Works)
V_LA_ (m/s)	3599	4000 [26]	3075 [22]	2300 [24]
V_TA_(m/s)	1086	1770 [26]	1010 [22]	1330 [24]
*C*_11_ (GPa)	41.0	39.5 [12]	35.9 [22]	32.3 [28]
*C*_44_ (GPa)	3.74	2.9 [12]	3.9 [22]	4.8 [28]

**Table 2 materials-15-03692-t002:** Fitting parameters for the relaxation time estimated by using Equation (4).

Fitting Parameter	*T* < −100 °C	*T* > −100 °C
$\tau_{0}$ (ps)	8.07	5.92
$T_{0}$ (°C)	−74.2 (199 K)	−133.0 (140 K)

## Data Availability

Data presented in this article is available on request from the corresponding author.

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
