# Peer review of "Acoustic Anomalies and the Critical Slowing-Down Behavior of MAPbCl3 Single Crystals Studied by Brillouin Light Scattering"

_materials, 2022, doi:10.3390/ma15103692_

Round 1

Reviewer 1 Report

Hybrid organic-inorganic halide perovskites are key materials for multiple applications. Elastic properties of halide perovskites are important for the investigation of their stability in device applications or material preparation. In this study, Lee et al investigated the temperature dependence of acoustic properties of MAP-bCl3 single crystals in the hypersonic range by using Brillouin spectroscopy. Finding results are interesting for a broad range of research area. The manuscript is well organized and the conclusion can be well supported by the results. Overall, I suggest it can be accepted for publication in this journal in its current form.

Author Response

Thank you very much for the reviewer’s favorable opinion on our manuscript. 

Reviewer 2 Report

In the manuscript, the authors investigated the acoustic properties of MAPbCl3 perovskite single crystals in a wide temperature range (-190 - 40 degrees Celsius) using Brillouin spectroscopy. They have calculated and analysed the temperature dependence of the sound velocity, the absorption coefficient, the optical constant (C11, C44) and the relaxation time of the modes TA and LA in a wide temperature range. Special attention was paid to the influence of phase transformation on the acoustic properties.

The results are clearly presented and discussed in detail. Furthermore, the authors have compared the obtained results with data from the literature and in this way strongly supported the discussion of the results.

There are some minor suggestions and comments:

  • Fig. 2A: Using only lines instead of lines and symbols would improve the clarity of the plot.

  • Fig. 2: The Brillouin spectra were taken during cooling. Did the authors check the spectra during the heating phase to confirm or find differences compared to cooling.

  • Fig. 3: The legend of the plot is somehow incomplete. symbols below -116 degrees Celsius should be described in the legend or in the caption.

  • For the sake of completeness, I suggest at least showing the FWHM for the TA mode in the appendix, especially since this is mentioned in the text.

  • Fig. 7: Please indicate the temperature for which the data are available (I assume room temperature).

Author Response

1. Fig. 2A: Using only lines instead of lines and symbols would improve the clarity of the plot.

  • Thank you very much for the reviewer’s favorable opinion on our manuscript. We have replaced the Fig. 2A on page 5 with updated version including only lines instead of lines and symbols.

2. Fig. 2: The Brillouin spectra were taken during cooling. Did the authors check the spectra during the heating phase to confirm or find differences compared to cooling.

  • We investigated the Brillouin spectra on heating with a larger temperature interval for roughly checking the acoustic properties on heating. From this process, we could confirm the splitting of the LA mode at low temperatures, the softening of the LA mode in addition to a significant damping factor in the cubic phase although the data are a little bit scattered, which may probably be due to the sample degradation caused by long exposure to the ambient condition. We included the following sentence in the manuscript “The Brillouin spectra were recorded upon heating as well. The comparison between cooling and heating processes is shown in Figure S2 in the Supplementary Materials. The splitting of the LA mode at low temperatures, the softening of the LA mode in addition to a significant damping factor in the cubic phase were confirmed in the heating process. However, the data are a little bit scattered, which may probably be due to the sample degradation caused by long exposure to the ambient condition.” (page 6)

3. Fig. 3: The legend of the plot is somehow incomplete. symbols below -116 degrees Celsius should be described in the legend or in the caption.

  • We replaced the figure 3 on page 6 with updated version in which we completely removed the legend to avoid confusion and described the legends in the figure caption. The caption was changed as “Temperature dependences of (a) the Brillouin frequency shift and (b) the FWHM of the LA mode propagating along the [100] direction measured upon cooling. The half-filled circles represent the split peaks”

4. For the sake of completeness, I suggest at least showing the FWHM for the TA mode in the appendix, especially since this is mentioned in the text.

  • We added the FWHM of the TA mode in the supplementary data as Figure S3.

5. Fig. 7: Please indicate the temperature for which the data are available (I assume room temperature).

  • For fig.7 on page 11, the temperature for the data was written in the figure caption. The caption was changed as “ Comparison of two elastic constants, C11 and C44, and the bulk modulus of three MA-based halide perovskites at room temperature. The C11 and C44 of MAPbCl3 are from the present work and other values are from several references as shown in Table 1.”

Reviewer 3 Report

The article studies the inelastic scattering of a single perovskite crystal by analyzing the behavior of the modes even when the temperature varies. The data show an anomalous behavior in the acoustic properties that is attributed to the coupling of the acoustic waves to the dynamics of the MA cations. The results are interesting and well described and can help in understanding the behavior of these interesting materials.

Author Response

(The authors gave the same response as above.)

Reviewer 4 Report

REFEREE REPORT

on paper Acoustic anomalies and a critical-slowing down behavior of MAPbCl3 single crystals studied by Brillouin light scattering

by authors Jeong Woo Lee, Furqanul Hassan Naqvi, Jae-Hyeon Ko, Tae Heon Kim and Chang Won Ahn,

submitted to Materials

The paper “Acoustic anomalies and a critical-slowing down behavior of MAPbCl3 single crystals studied by Brillouin light scattering” is devoted to preparation and investigation of organic-inorganic halide perovskite MAPbCl3 single crystals. X-ray diffraction, Brillouin spectroscopy in a wide temperature range, absorption coefficient were applied and examined. The topic of this paper is critically actual especially in optoelectronics and photovoltaics. The data are reliable and do not cause much doubt. Nevertheless, there are several points before the paper can be published. I hope that authors after minor revisions can improve the paper and can publish it in Materials.

  1. I believe that it is not necessary to include PACS numbers in the article.
  2. Abstract sound to general, do not stand alone with the paper and do not describe the main results of the manuscript. It should be revised.
  3. The Introduction part must be improved with literature in the field of perovskites and I suggest to use the following reference (see and discuss:

https://doi.org/10.1016/j.ceramint.2021.12.110;

https://doi.org/10.4028/www.scientific.net/SSP.299.100).

  1. Introduction part should be improved with the information about other technologies of perovskites preparation.
  2. XRD results should be placed in Results and Discussion part. The information about diffractometer used should be added in Experimental part.
  3. Why did you choose the following temperature range +40…-190ºÐ¡? Why did you not heat samples more than 40ºÐ¡?
  4. The use of superscripts for the reference numbers indicating in Table 1 is confusing. Please use square brackets to indicate the references.
  5. Conclusion part is too short, please improve it.
  6. There are some insufficient typos and English mistakes in the text.

But any way I impressed by this paper. But authors must explain some details and improve the paper in accordance with my comments. The paper should be sent to me for the second analysis after the minor revisions.

Author Response

1. I believe that it is not necessary to include PACS numbers in the article.

  • Thank you very much for the reviewer’s favorable opinion on our manuscript. PACS numbers were removed from the manuscript.

2. Abstract sound to general, do not stand alone with the paper and do not describe the main results of the manuscript. It should be revised.

  • We cannot understand exactly what this comment indicates. All sentences in the abstract are directly related to the major, specific findings of the present study, and we tried to summarize them in a compact manner. If the reviewer asks us to modify the abstract in a more general direction not focusing on specific findings, we will do that accordingly.

3. The Introduction part must be improved with literature in the field of perovskites and I suggest to use the following reference (see and discuss:

https://doi.org/10.1016/j.ceramint.2021.12.110;

https://doi.org/10.4028/www.scientific.net/SSP.299.100).

  • The introduction was revised by improving literature. The references suggested were added by adding following new line in the beginning of paragraph 1 at page 1. [1,2] “Perovskites with a generic ABX3 structure are being employed in several research fields[1,2]”

4. Introduction part should be improved with the information about other technologies of perovskites preparation.

  • The introduction was improved by adding more information about other technologies of perovskite preparation.[3] New lines at the end of paragraph 1 at page 1 were added. Single crystals can be made using a variety of techniques, including solvent evaporation, temperature lowering, inverse crystallization, and antisolvent vapor assisted methods [10]. We chose the solvent evaporation method among these to synthesize the single crystals.

5. XRD results should be placed in Results and Discussion part. The information about diffractometer used should be added in Experimental part.

  • The last line of second paragraph at page 3 was removed and transferred to first paragraph of Results and Discussion section.
  • Secondly, information about diffractometer used was added in Experimental part as new first paragraph on page 4: “A high-resolution powder X-ray diffractometer (PANalytical X’pert Pro MPD) was used to record X-ray diffraction (XRD) patterns at room temperature, with Cu Kβ radiations with λ =1.5406 Å as excitation source to probe the crystal at a scan rate of 3° per minute in a scan range of 10° < 2Θ < 60°.”

6. Why did you choose the following temperature range +40…-190ºÐ¡? Why did you not heat samples more than 40ºÐ¡?

  • In this study, the investigated temperature range is limited from the lowest temperature (-196°C) to near room temperature (40°C). Generally, above room temperature, this material does not show any structural phase transitions except for thermal degradation. Therefore, the behavior of the material above room temperature is out of scope of this study.

7. The use of superscripts for the reference numbers indicating in Table 1 is confusing. Please use square brackets to indicate the references. 

  • Square brackets were used to indicate the references. 

8.  Conclusion part is too short, please improve it.

  • We modified the conclusion to describe more details of our findings. (page 17)

9. There are some insufficient typos and English mistakes in the text.

  • Thank you for your suggestion. The paper was carefully scrutinized and several typos and English mistakes were corrected.
